# Anticonvulsant Potential and Toxicological Profile of *Verbesina persicifolia* Leaf Extracts: Evaluation in Zebrafish Seizure and *Artemia salina* Toxicity Models

**DOI:** 10.3390/plants14071078

**Published:** 2025-04-01

**Authors:** Carlos Alberto López-Rosas, Santiago González-Periañez, Tushar Janardan Pawar, Jorge Iván Zurutuza-Lorméndez, Fernando Rafael Ramos-Morales, José Luís Olivares-Romero, Margarita Virginia Saavedra Vélez, Fabiola Hernández-Rosas

**Affiliations:** 1Instituto de Química Aplicada, Universidad Veracruzana, Luis Castelazo Ayala s/n, Col. Industrial Animas, Xalapa 91190, Mexico; carloslopez02@uv.mx (C.A.L.-R.); santiagonzalez@uv.mx (S.G.-P.); framos@uv.mx (F.R.R.-M.); 2Instituto de Neuroetología, Universidad Veracruzana, Luis Castelazo Ayala s/n, Col. Industrial Animas, Xalapa 91190, Mexico; 3Facultad de Química Farmacéutica Biológica, Universidad Veracruzana, Circuito Aguirre Beltrán s/n, Col. Zona UV, Xalapa 91090, Mexico; 4Red de Estudios Moleculares Avanzados, Campus III, Instituto de Ecología A. C., Carretera Antigua a Coatepec 351, Xalapa 91073, Mexico; tushar.janardan@inecol.mx (T.J.P.); jose.olivares@inecol.mx (J.L.O.-R.); 5Centro de Salud Urbano José A. Maraboto Carreón, Servicios de Salud de Veracruz, Santiago Bonilla No 85, Col. Obrero Campesino, Xalapa 91020, Mexico; zurutuza1111@hotmail.com; 6Centro de Investigación, Universidad Anahuac Querétaro, El Marqués, Querétaro 76246, Mexico; 7Escuela de Ingeniería Biomédica, División de Ingenierías, Universidad Anahuac Querétaro, El Marqués, Querétaro 76246, Mexico; 8Facultad de Química, Universidad Autónoma de Querétaro, Querétaro 76010, Mexico

**Keywords:** *Verbesina persicifolia*, anticonvulsant activity, zebrafish epilepsy model, PTZ-induced seizures, *Artemia salina* toxicity, GABAergic modulation, flavonoids, alkaloids, diazepam synergy

## Abstract

Epilepsy is a chronic neurological disorder with significant treatment challenges, necessitating the search for alternative therapies. This study evaluates the anticonvulsant activity and toxicological profile of *Verbesina persicifolia* leaf extracts. Methanolic and sequential fractions (hexane, dichloromethane, ethyl acetate, and methanol) were tested using a pentylenetetrazole (PTZ)-induced seizure model in zebrafish (*Danio rerio*), measuring seizure latency, severity, and survival rates. Phytochemical screening confirmed the presence of flavonoids, alkaloids, and steroids, suggesting potential neuroactive properties. The hexane extracts significantly increased seizure latency and survival rates, with co-administration of hexane extract (5 µg/mL) and diazepam (35.5 µM) further enhancing these effects. Toxicity assessment in *Artemia salina* indicated low to moderate toxicity in methanolic extracts, while sequential fractions exhibited higher toxicity, particularly in hexane and ethyl acetate extracts. These findings suggest that *V. persicifolia* extracts exert anticonvulsant effects, likely through GABAergic modulation, and exhibit a favorable safety profile at therapeutic doses. The results support further investigations to isolate active constituents, confirm their mechanisms of action, and explore their potential as plant-derived anticonvulsant agents.

## 1. Introduction

Epilepsy is a prevalent neurological disorder affecting approximately 50 million people worldwide [1]. It is characterized by recurrent and unprovoked seizures caused by abnormal neuronal activity in the brain [2]. The disorder significantly impacts patients’ quality of life, contributing to increased morbidity, mortality, psychological distress, and social stigmatization. Global epidemiological studies indicate an incidence of epilepsy ranging from 50.4 to 81.7 per 100,000 people per year, with the highest prevalence in children and older adults [3]. In Mexico, epilepsy is a leading cause of neurological-related deaths, with 1555 reported fatalities in 2023, predominantly affecting individuals between the ages of 1 and 14 years Additionally, 6634 patients required hospitalization due to epilepsy-related complications, underscoring the disease’s burden on public health systems [4,5]. In many low- and middle-income countries (LMICs), up to 70% of epilepsy patients lack access to appropriate treatment, contributing to higher mortality rates and disability-adjusted life years (DALYs). By contrast, high-income countries benefit from greater healthcare access, reducing mortality and improving disease management. Furthermore, epilepsy places a significant economic burden on healthcare systems, with direct costs related to hospitalizations, medications, and specialist care, as well as indirect costs from loss of productivity and social stigma. These disparities further emphasize the urgent need for alternative, accessible, and cost-effective treatment strategies, particularly in resource-limited settings [6,7].

Current antiepileptic drugs (AEDs) primarily function by modulating ion channels, enhancing inhibitory neurotransmission via the GABAergic system, or reducing excitatory glutamatergic signaling. However, approximately 30% of patients develop drug-resistant epilepsy, meaning their seizures remain uncontrolled despite pharmacological intervention [8]. Moreover, AEDs often cause significant adverse effects, including cognitive impairment, behavioral changes, systemic toxicity, teratogenicity, and hypersensitivity reactions, all of which reduce patient adherence and overall quality of life [9,10]. These limitations highlight the urgent need for novel therapeutic options with improved efficacy, better safety profiles, and fewer side effects.

Given these challenges, natural compounds have gained increasing interest in epilepsy research. Medicinal plants have historically contributed to drug discovery, with numerous plant-derived molecules being successfully developed into modern pharmaceuticals. For example, galantamine, originally isolated from the Galanthus species, is now widely used to treat Alzheimer’s disease, illustrating the pharmacological potential of plant-derived compounds [11]. Various phytochemicals, such as flavonoids, alkaloids, terpenoids, and phenolics, have demonstrated neuroprotective, anticonvulsant, anxiolytic, and antidepressant effects, making them promising candidates for epilepsy management [12,13,14]. Furthermore, multiple natural products, including curcumin, resveratrol, and berberine, have exhibited anticonvulsant activity through mechanisms involving oxidative stress modulation, neuroinflammation reduction, and neurotransmitter regulation. Despite this growing research, many medicinal plants remain unexplored for their potential applications in epilepsy treatment [15,16,17].

The Verbesina genus, part of the Asteraceae family, has been traditionally used in herbal medicine for a variety of therapeutic purposes. Some species within this genus have demonstrated anti-inflammatory, antimicrobial, antioxidant, and neuroactive effects, suggesting their potential for neurological applications [18]. However, despite these promising properties, the neuropharmacological potential of *Verbesina persicifolia* remains largely unstudied. While preliminary reports suggest that some Verbesina species may exhibit anxiolytic and antidepressant effects, no published studies have investigated the anticonvulsant potential of *V. persicifolia* [19,20,21] This knowledge gap represents a critical research opportunity to evaluate the plant’s bioactive compounds and their potential role in epilepsy treatment. Given that previous studies have suggested that other members of the Verbesina genus possess neuroactive properties, it is plausible that *V. persicifolia* contains bioactive compounds capable of influencing seizure susceptibility.

The zebrafish (*Danio rerio*) model has emerged as a powerful tool in neurological and epilepsy research, offering unique advantages due to its high genetic and neurophysiological similarity to mammals. Its small size, optical transparency, and well-characterized behavior allow for efficient, high-throughput screening of neuroactive compounds [22]. In epilepsy studies, zebrafish are widely used to model seizure disorders induced by pharmacological agents such as pentylenetetrazole (PTZ), a GABA receptor antagonist. PTZ is known to induce seizure-like activity by disrupting inhibitory neurotransmission, making it an established tool for evaluating anticonvulsant drug candidates [23]. The zebrafish PTZ model reliably mimics human seizure mechanisms, making it a valuable platform for testing potential antiepileptic agents [24,25].

In this study, we investigate the anticonvulsant potential of *V. persicifolia* extracts using a zebrafish PTZ-induced seizure model. Extracts were obtained using sequential and split extraction methods to maximize the identification of potential bioactive compounds. A preliminary phytochemical analysis was conducted to screen neuroactive secondary metabolites that may contribute to the plant’s pharmacological effects. By integrating ethnopharmacological knowledge with experimental neuropharmacology, this research provides insights into the potential anticonvulsant properties of *V. persicifolia*, contributing to the broader search for plant-derived therapies for epilepsy.

## 2. Results

The extraction of *Verbesina persicifolia* leaves yielded hexane, dichloromethane, ethyl acetate, and methanol fractions, which were analyzed for phytochemical composition, toxicity, and anticonvulsant activity. Bioactive compounds were identified in the extracts, and toxicity evaluation in *Artemia salina* was conducted alongside anticonvulsant assessment in a PTZ-induced zebrafish seizure model.

### 2.1. Extraction and Phytochemical Analysis of Verbesina persicifolia

The phytochemical composition of *Verbesina persicifolia* leaf extracts was analyzed for three consecutive years (2019, 2020, and 2021). Phytochemical screening of the extracts confirmed the presence of alkaloids, flavonoids, and steroids in all years, while saponins, tannins, quinones, and triterpenoids showed variability between collection years (Table 1). These differences may be influenced by environmental factors affecting the biosynthesis of secondary metabolites (Appendix A). It should be noted that for the anticonvulsant activity, extracts obtained from the 2021 collection were used.

To complement our findings, previous phytochemical studies on *Verbesina persicifolia* and related *Verbesina* species have identified eudesmane sesquiterpenes, terpenoids, flavonoids, alkaloids, and essential oils. Table 2 summarizes relevant secondary metabolites previously reported in the genus *Verbesina* and *Verbesina persicifolia* [18,26,27].

Thin-layer chromatography (TLC) was performed to confirm the presence of these metabolites in the hexane, dichloromethane, ethyl acetate, and methanol extracts. The analysis revealed distinct bands corresponding to flavonoids, phenols, steroids, and alkaloids, with optimal separation achieved using dichloromethane/methanol (95:5) for non-polar fractions and ethyl acetate/methanol (8:2) for polar fractions. However, due to the low quality of the TLC images, these figures have been provided as Appendix A.

The ^1^H NMR spectrum of the *Verbesina persicifolia* extracts (hexane, dichloromethane, and ethyl acetate) displayed characteristic signals indicative of both flavonoid derivatives and eudesmane sesquiterpenes (Appendix A). In the aromatic region (7.6–7.4 ppm and 6.4–6.5 ppm), multiple proton signals suggest the presence of hydroxylated or methoxylated flavonols, while multiplets in the glycosidic region (3.0–5.5 ppm) indicate possible flavonoid glycosides, such as rutin or quercetin derivatives.

Additionally, methyl signals in the aliphatic region (0.9–1.5 ppm) confirmed the presence of eudesmane sesquiterpenes, which are consistent with previously reported sesquiterpene lactones in *Verbesina* species [26,28,29,30,31,32]. While the NMR data strongly support the co-occurrence of flavonoids and sesquiterpenes, the identification of rutin or specific glycosylated flavonoids remains inconclusive due to the lack of complete characterization. Further structural confirmation via advanced NMR techniques and mass spectrometry is required to precisely characterize these bioactive constituents.

### 2.2. Toxicological Assessment in Artemia salina

The toxicity of *Verbesina persicifolia* extracts was evaluated using the median lethal concentration (LC_50_) values, classified according to Clarkson’s toxicity scale. Extracts with LC_50_ > 1 mg/mL were considered non-toxic, while those within 0.5–1 mg/mL, 0.1–0.5 mg/mL, and <0.1 mg/mL were classified as low, moderate, and high toxicity, respectively.

The methanolic partitions exhibited low- to non-toxic profiles, with the hexane and ethyl acetate fractions showing low toxicity (LC_50_ values of 0.499 mg/mL and 0.633 mg/mL, respectively), while the dichloromethane and methanol fractions were non-toxic (LC_50_ > 1 mg/mL) (Table 3).

In contrast, the sequentially extracted fractions demonstrated increased toxicity, particularly in the hexane (LC_50_ = 0.073 mg/mL, high toxicity) and dichloromethane (LC_50_ = 0.105 mg/mL, moderate toxicity) extracts. Both the ethyl acetate and methanol sequential extracts exhibited moderate toxicity (LC_50_ = 0.421 mg/mL) (Table 3).

These findings confirm that the methanolic partitions of *Verbesina persicifolia* demonstrate a generally safe profile, whereas sequentially extracted fractions exhibit greater toxicity, particularly in non-polar extracts.

### 2.3. Anticonvulsant Activity in Zebrafish Model

The anticonvulsant potential of *Verbesina persicifolia* extracts was assessed in PTZ-induced seizures in zebrafish. The key parameters measured included Latency IV, Whirlpool Latency, Posture Loss Latency, and survival rates. Extracts were evaluated using partition and sequential extraction methods, with the hexane fraction also tested in combination with diazepam and sodium valproate.

#### 2.3.1. Partition Method

The partition method was used to investigate the anticonvulsant effects of different solvent extracts (hexane, dichloromethane, ethyl acetate, methanol, and water).

**Speed Index Latency**: The different state extracts had no significant differences between the groups and the control group (Figure 1A).

**Whirlpool Latency**: The dichloromethane extract at 1000 µg/mL significantly prolonged Whirlpool Latency compared to the control (*p* ≤ 0.01) (Figure 1B).

**Posture Loss Latency**: The dichloromethane extract at 1000 µg/mL had the highest latency (402.33 s), significantly differing from the control (*p* = 0.03). However, higher concentrations correlated with reduced survival rates (Figure 1C).

#### 2.3.2. Sequential Method

The sequential extraction method was applied to further evaluate anticonvulsant activity using the hexane, dichloromethane, ethyl acetate, methanol, and water fractions.

**Speed Index Latency:** The ethyl acetate extract had the strongest effect, with 10 and 100 µg/mL producing latencies of 334.67 and 486.67 s, respectively; however, there were no significant differences with the control (Figure 2A).

**Whirlpool Latency:** The ethyl acetate extract at 10 and 100 µg/mL significantly increased latency times (892.33 and 732.33 s, respectively). Both concentrations differed significantly from the control and lower doses (*p* ≤ 0.01). Hexane and dichloromethane extracts did not show significant effects (Figure 2B).

**Posture Loss Latency:** The ethyl acetate extract at 10 and 100 µg/mL exhibited the longest latencies, indicating a dose-dependent effect (<0.01). Other extracts did not produce significant differences across concentrations (Figure 2C).

### 2.4. Co-Administration with Pharmacological Controls

The hexane partition extract of *Verbesina persicifolia* was co-administered with the standard anticonvulsant drugs diazepam (Dzp) and sodium valproate (Val-Na) to evaluate potential synergistic effects. The behavioral parameters recorded included Latency IV, Whirlpool Latency, and Posture Loss Latency.

**Speed Index Latency:** No statistically significant differences were observed between the control, pharmacological controls, and co-administration groups. However, the hexane + diazepam combination exhibited a slight, though non-significant, increase in latency compared to diazepam alone, suggesting a possible additive effect (Figure 3A).

**Whirlpool Latency:** Significant differences were observed between diazepam, the hexanic fraction of 15 µg/mL, and the co-administration of diazepam with the hexanic fraction (*p* =< 0.01) (Figure 3B).

**Posture Loss Latency:** Co-administration of the hexane extract with diazepam significantly increased Posture Loss Latency, whereas co-administration with sodium valproate did not yield a comparable effect. These findings indicate that the diazepam–hexane extract combination may provide a greater anticonvulsant effect than sodium valproate (Figure 3C). The control had significant differences against the diazepam and hexane partition of 15 µg/mL groups, which had a significance of *p* =< 0.01. On the other hand, the control against sodium valproate and the co-administration of diazepam and the hexane partition had a significance of *p* = 0.01. In this sense, co-administration of the hexanic fraction may be an adjuvant in the treatment of seizures.

### 2.5. Survival Rate

The survival rates of zebrafish exposed to *Verbesina persicifolia* extracts were analyzed across different experimental conditions (Figure 4).

**Partition Method:** The hexane extract showed the highest survival rate, reaching 100% at 1000 µg/mL. The dichloromethane extract exhibited a dose-dependent increase, with survival reaching 65% at 100 and 1000 µg/mL. In contrast, the ethyl acetate extract maintained a stable survival rate of approximately 85% across all concentrations (Figure 4A).

**Sequential method:** The hexane extract attained 100% survival at 100 µg/mL, whereas other extracts, such as methanol, exhibited lower survival rates, with a maximum of 60% across all concentrations (Figure 4B).

**Co-administration assays:** Diazepam, sodium valproate, and hexane extract (including co-administration treatments) achieved 100% survival, while the control group exhibited only 14.28% survival.

## 3. Discussion

This study demonstrates the anticonvulsant potential of *Verbesina persicifolia* leaf extracts, particularly the ethyl acetate and hexane fractions, in a PTZ-induced zebrafish seizure model. The results suggest that key phytochemicals, including flavonoids, alkaloids, and steroids, play a significant role in seizure modulation. This section discusses the implications of these findings, potential mechanisms of action, and the relevance of *Verbesina persicifolia* as a candidate for novel anticonvulsant development.

### 3.1. Phytochemical Contributions to Anticonvulsant Activity

Phytochemical screening confirmed the presence of flavonoids, alkaloids, and steroids in *Verbesina persicifolia* extracts, all of which have well-documented neuropharmacological properties. Flavonoids are known to interact with γ-aminobutyric acid (GABA) receptors, enhancing GABAergic transmission, which may contribute to the anticonvulsant effects observed in the ethyl acetate fraction [33,34,35,36,37]. Similarly, alkaloids have been reported to modulate neurotransmitter systems, either by acting as GABA agonists or by inhibiting excitatory pathways, further supporting their role in seizure suppression [38,39,40]. Steroids have also been implicated in central nervous system excitability, suggesting their possible contribution to the observed anticonvulsant activity.

The increases in convulsion latency and survival rates observed in this study suggest that these phytochemicals contribute to both the enhancement of inhibitory neurotransmission and the reduction of excitatory signaling. This aligns with established models of seizure modulation, where balancing excitatory and inhibitory pathways is crucial for effective seizure control. These findings are consistent with studies on other *Verbesina* species, such as *Verbesina encelioides* [41] and *Verbesina crocata* [42], which contain related bioactive compounds with neuroactive properties. The presence of similar metabolites in *Verbesina persicifolia* reinforces its potential as a candidate for further anticonvulsant research.

### 3.2. Efficacy of Extracts in Reducing Seizure Parameters

The ethyl acetate fraction showed the most substantial effect, delaying convulsion onset and reducing severe seizure behaviors. This fraction likely contains non-polar flavonoids and other bioactive compounds, which may improve blood–brain barrier permeability, enhancing CNS modulation.

The hexane fraction also demonstrated potent anticonvulsant effects, achieving 100% survival at 10 µg/mL, further supporting its therapeutic potential. This aligns with previous findings indicating that non-polar compounds exhibit stronger neuroactive effects, likely due to higher bioavailability in the CNS [43].

Something important to highlight is that although the dichloromethane and ethyl acetate fractions increased the evaluated latencies, they had a certain percentage of mortality; in contrast, the hexane fraction, although it did not have a marked reduction in the evaluated variables, did have a 100% survival rate at the concentration of 10 µg/mL.

Together, these data suggest that the seizure-modulating activity of *Verbesina persicifolia* fractions results from a combination of inhibitory neurotransmission enhancement and excitatory pathway suppression. This aligns with established seizure control mechanisms, making these extracts promising candidates for further research in anticonvulsant drug development.

### 3.3. Synergistic Effects in Co-Administration with Standard Anticonvulsants

The co-administration experiments demonstrated that the hexane fraction of *Verbesina persicifolia* significantly enhanced seizure latency and survival rates when combined with diazepam, suggesting a potential synergistic interaction. Diazepam is a well-established anticonvulsant that potentiates GABAergic activity by binding to GABA_A_ receptors, increasing inhibitory neurotransmission and reducing seizure susceptibility [44]. The observed enhancement in anticonvulsant effects when *Verbesina persicifolia* hexane extract was co-administered with diazepam suggests that compounds within the extract may either further potentiate GABA_A_ receptor activity or modulate additional pathways involved in seizure control.

Compared to sodium valproate, the combination of the hexane extract and diazepam resulted in a nearly threefold increase in seizure latency, highlighting a greater efficacy in delaying convulsions. This effect suggests that the hexane fraction may contain bioactive compounds capable of amplifying GABAergic signaling, reducing neuronal excitability, or enhancing diazepam’s action at lower doses. The ability to use lower doses of standard anticonvulsants while maintaining efficacy is particularly relevant in epilepsy management, as it could help mitigate the side effects associated with long-term use of conventional drugs.

Similar synergistic effects have been observed in other plant-derived compounds, where phytochemicals enhance the therapeutic action of conventional drugs by either increasing bioavailability, potentiating receptor interactions, or modulating secondary signaling pathways. The observed interaction between *Verbesina persicifolia* extracts and diazepam aligns with these findings, reinforcing the potential of plant-derived compounds as adjunct therapies in epilepsy management. Further research is needed to isolate the specific active constituents responsible for this synergy and to elucidate their precise mechanisms of action within the central nervous system.

### 3.4. Toxicity and Safety Profile

Toxicity evaluation in *Artemia salina* revealed that the methanolic fractions exhibited low to moderate toxicity, aligning with established safety thresholds for medicinal plant extracts. However, the sequentially extracted fractions (hexane and ethyl acetate) exhibited higher toxicity, likely due to increased concentrations of secondary metabolites such as terpenes and alkaloids, which are known to be cytotoxic at higher doses.

### 3.5. Implications and Future Directions

The promising results from this study reinforce the potential of *Verbesina persicifolia* as a therapeutic agent for seizure management, particularly when used in combination with standard anticonvulsants. The observed anticonvulsant activity of its hexane and ethyl acetate fractions suggests the presence of bioactive compounds that modulate seizure pathways, likely through GABAergic enhancement and inhibition of excitatory neurotransmission. These findings contribute to the growing body of evidence supporting the role of medicinal plants in neurological disorders and highlight the importance of further research to fully characterize their pharmacological properties.

The main limitation of the study was that crude extracts were used which were not fractioned, which does not allow us to identify any compound(s) responsible for the anticonvulsant activity. Future studies should focus on the isolation and structural characterization of the active constituents responsible for the anticonvulsant effects observed in this study. Advanced analytical techniques, such as high-performance liquid chromatography (HPLC) and nuclear magnetic resonance (NMR) spectroscopy, could be employed to identify specific flavonoids, alkaloids, and other secondary metabolites involved in seizure modulation. Additionally, mechanistic studies should investigate how these compounds interact with neurotransmitter receptors and ion channels within the central nervous system to better understand their mode of action.

Regarding data and statistical analysis, increasing the sample size and reducing the study groups is essential to avoid committing a type 2 error. This initial approach allowed us to identify the most relevant group to optimize the study design, using only those groups that showed a statistically proven effect compared to the control group. Furthermore, the mortality rate analysis selects those compounds with an effect and no mortality risk.

Beyond phytochemical characterization, in vivo studies using rodent models could provide further insights into the long-term efficacy, safety, and pharmacokinetics of *Verbesina persicifolia* extracts. Evaluating different dosages and administration routes will be critical in determining their clinical potential. Furthermore, exploring the potential for *Verbesina persicifolia* as an adjunct therapy to conventional antiepileptic drugs could offer new treatment options, particularly for drug-resistant epilepsy cases.

The increasing interest in plant-based therapies, especially in regions with limited access to conventional epilepsy treatments, underscores the importance of integrating traditional medicine with modern pharmacological approaches. Given its promising anticonvulsant activity and favorable safety profile at lower doses, *Verbesina persicifolia* represents a strong candidate for further drug development. Expanding research efforts in this area could lead to new, more accessible, and potentially safer alternatives for managing epilepsy and other neurological disorders.

## 4. Materials and Methods

### 4.1. Plant Material Collection

*Verbesina persicifolia* leaves were collected annually in September from 2019 to 2021 in Arroyo del Potrero, municipality of Martínez de la Torre, Veracruz, Mexico, at the coordinates 20°08′43.0″ N 97°03′56.1″ W. The collected material was carefully inspected to remove any damaged or yellowed leaves before processing. The leaves were air-dried in the shade with frequent turning to prevent excess moisture accumulation and potential degradation of secondary metabolites. Once fully dried, the plant material was finely ground using a mechanical grinder and stored under vacuum-sealed conditions to prevent oxidation and contamination before extraction.

For botanical authentication, a voucher specimen (Boucher *Verbesina persicifolia* D.C.-AAM-002-XAL) was deposited at the National Herbarium of the Instituto Nacional de Ecología (INECOL), Mexico, where its identity was confirmed. This ensured accurate taxonomic classification and consistency across different collection years.

### 4.2. Preparation of Extracts

Two extraction methods were employed to isolate bioactive compounds from *Verbesina persicifolia* leaves. In the sequential maceration method, 100 g of finely ground plant material was used per liter of solvent, which was macerated in hexane for 21 days in amber flasks, with decanting of the solvent every seven days. The solvent was removed under reduced pressure using a BUCHI R-210 rotary evaporator (Buchi AG, Flawil, Switzerland), and the same process was subsequently performed with dichloromethane, ethyl acetate, and methanol, resulting in four distinct solvent fractions.

For the partitioning method, an initial methanolic extract was prepared by macerating 350 g of plant material in methanol for 21 days. After filtration, the extract was concentrated under reduced pressure, and partitions were created using hexane, dichloromethane, ethyl acetate, and water. The solvents were supplied by Meyer (México) and were used with further purification. This method was adapted from [45,46] to optimize solvent fractionation and ensure efficient separation of bioactive metabolites.

### 4.3. Phytochemical Screening

The methanolic extracts from each collection year were subjected to phytochemical screening to identify the presence of secondary metabolite families such as alkaloids, saponins, tannins, flavonoids, and steroids. This analysis was based on colorimetric reactions [47].

^1^H NMR spectra were recorded on a 500 MHz NMR spectrometer using CDCl_3_ as the solvent and tetramethylsilane (TMS) as an internal standard. The solvent was supplied by Sigma-Aldrich (Saint Louis, MO, USA). Chemical shifts (δ) are reported in ppm, and coupling constants (J) are given in Hz. Spectra were acquired for the hexane, dichloromethane, and ethyl acetate extracts to analyze the chemical composition of *Verbesina persicifolia*. The obtained spectral data were compared with previous reports on the *Verbesina* genus to aid in the identification of major classes of natural products present in the extracts [26,30,32].

### 4.4. Toxicological Evaluation in Artemia salina

The toxicity of *Verbesina persicifolia* extracts was evaluated using the brine shrimp lethality assay (BSLA) with *Artemia salina* nauplii, following established methodologies [48,49]. Approximately 350 mg of *A. salina* cysts were hatched in 2 L of artificial seawater (3% NaCl solution) under controlled conditions, including constant aeration, temperature (24–29 °C), and continuous light exposure. After 48 h, the hatched nauplii were transferred to experimental test wells containing various concentrations of *Verbesina persicifolia* methanolic extracts (1, 0.5, 0.25, 0.1, and 0.05 mg/mL) prepared in 3% saline solution.

After 24 h of exposure, the number of surviving nauplii was recorded, and mortality rates were determined. The median lethal concentration (LC_50_) was calculated using Probit analysis with 95% confidence intervals, employing IBM SPSS Statistics 29 for statistical analysis.

### 4.5. Anticonvulsant Activity in Zebrafish Model

The anticonvulsant potential of *Verbesina persicifolia* extracts was evaluated in adult zebrafish (*Danio rerio*) using a pentylenetetrazol (PTZ)-induced seizure model (Sigma-Aldrich (St. Louis, MO, USA). Zebrafish were pre-treated by immersion in tanks containing different extract concentrations for 30 min before seizure induction. Methanolic extracts were tested at 1, 10, 100, and 1000 µg/mL, while partitioned fractions were evaluated at 0.1, 1, 10, and 100 µg/mL. 

Following pre-treatment, zebrafish were transferred to a 10 mM PTZ solution to induce convulsions. Behavioral responses were recorded and classified into three distinct seizure stages: Speed Index Latency, rapid movement in all directions; Whirlpool Latency, circular movements; and Posture Loss Latency, wild jumping, clonic-like movement (seizure), then loss of coordination/posture, remaining at the bottom. Survival rates were monitored post-exposure to assess the potential neuroprotective effects of the extracts. This protocol was adapted from [50,51].

### 4.6. Co-Administration with Pharmacological Controls

To evaluate potential synergistic anticonvulsant effects, the hexane fractions of *Verbesina persicifolia*, which demonstrated significant activity in the zebrafish seizure model, were co-administered with diazepam and sodium valproate. Both pharmacological controls were obtained from commercially available tablets and prepared by dissolving diazepam in methanol and sodium valproate in ethyl acetate to ensure solubility and bioavailability.

Following pre-treatment with *Verbesina persicifolia* hexane extracts, zebrafish were exposed to diazepam or sodium valproate, and key seizure parameters, including convulsion latency, seizure severity, and survival rates, were recorded. The experimental design followed methodologies adapted from [52].

### 4.7. Statistical Analysis

All data analyses were performed using IBM SPSS Statistics 29 and GraphPad Prism 8. Results were expressed as mean ± standard error of the mean (SEM). The median lethal concentration (LC_50_) values for *Artemia salina* toxicity assays were determined using Probit analysis, with 95% confidence intervals calculated to assess dose-dependent toxicity.

For seizure latency (Latency Speed Index, Whirlpool Latency, and Posture Loss Latency) in the zebrafish model, normality was tested using the Shapiro–Wilk test, and homogeneity of variance was assessed with Levene’s test. If assumptions of normality and homoscedasticity were met, parametric tests such as one-way ANOVA followed by Tukey HSD as a post hoc test were used for multiple comparisons. In cases where data did not meet normality assumptions, non-parametric alternatives, such as the Kruskal–Wallis test with Bonferroni correction for the Mann–Whitney U test used as post hoc correction, were applied. Statistical significance was set at *p* < 0.05 for all analyses. For the construction of the comparison graphs, due to the small sample size and high variability, a boxplot was chosen for the best representation of the data, where the median and the 25th and 75th percentiles are represented, since the mean and standard deviation could be biased.

## 5. Conclusions

This study demonstrates the anticonvulsant potential of *Verbesina persicifolia* leaf extracts, particularly the ethyl acetate and hexane fractions, which significantly prolonged seizure latency and improved survival rates in a PTZ-induced zebrafish model. Phytochemical screening confirmed the presence of flavonoids and alkaloids, suggesting their role in GABAergic modulation and seizure suppression. Notably, the hexane fraction exhibited a synergistic effect with diazepam, indicating its potential as an adjunct therapy to conventional anticonvulsants, which may allow for dose reduction and minimize side effects. Toxicological evaluation in *Artemia salina* confirmed a favorable safety profile at therapeutic concentrations, although higher doses, particularly in non-polar fractions, showed increased toxicity.

Despite these promising findings, the study has some limitations, including the use of crude extracts, which prevents precise identification of active anticonvulsant compounds, and batch variations in plant material affecting metabolite composition. Additionally, the sample size in some experiments was limited, which could influence the statistical power. Future research should focus on isolating and characterizing bioactive compounds, elucidating their mechanisms of action, and validating efficacy in rodent epilepsy models. Investigating the long-term safety profile and pharmacokinetics of *Verbesina persicifolia* extracts will be crucial for potential therapeutic development.

## Figures and Tables

**Figure 1 plants-14-01078-f001:**
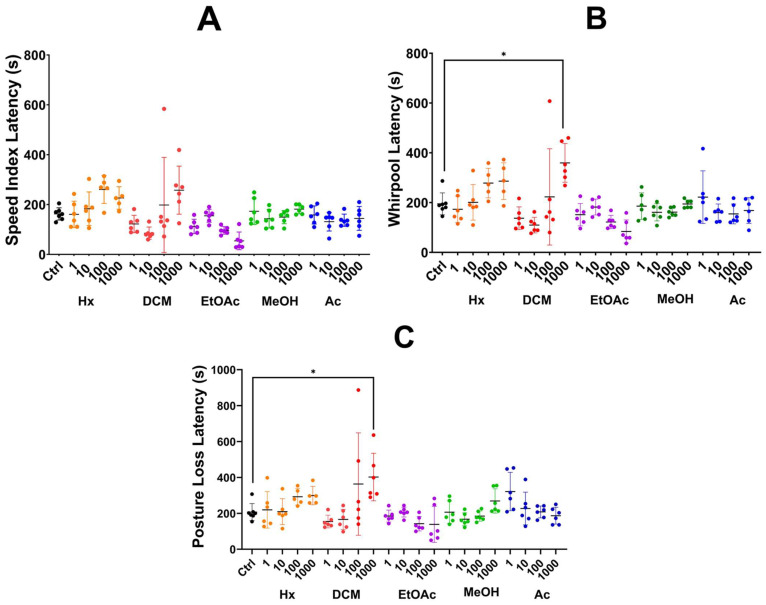
Latency metrics for zebrafish treated with *Verbesina persicifolia* extracts obtained by the partition method. (**A**) No significant differences; (**B**) (* *p* ≤ 0.01); (**C**) (* *p* = 0.03). Data are presented as medians ± interquartile ranges. The analysis was performed using ANOVA followed by Tukey HSD.

**Figure 2 plants-14-01078-f002:**
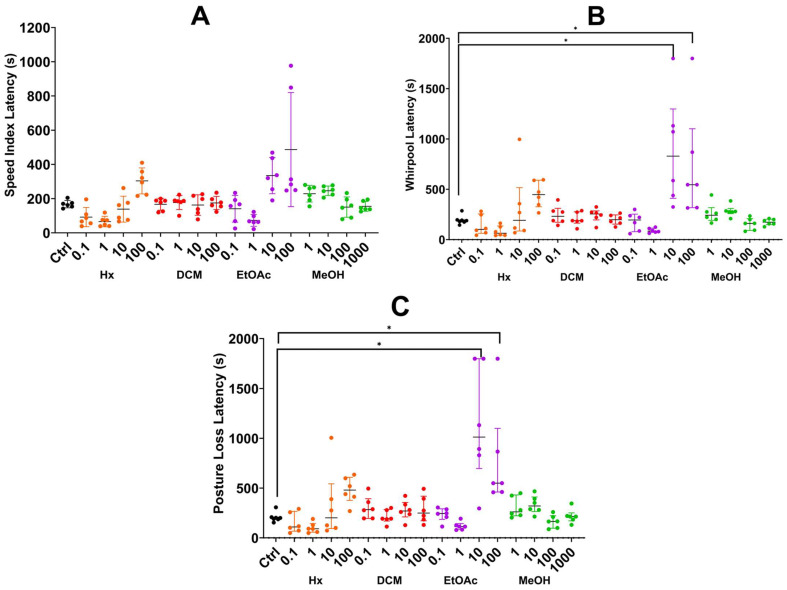
Latency metrics for zebrafish treated with *Verbesina persicifolia* extracts obtained by the sequential method. (**A**) Speed Index Latency, (**B**) Whirlpool Latency, (**C**) Posture Loss Latency. (**B**) (* *p* = < 0.01); (**C**) (* *p* = < 0.01). Data are presented as medians +- interquartile ranges. The analysis was performed using ANOVA followed by Tukey HSD.

**Figure 3 plants-14-01078-f003:**
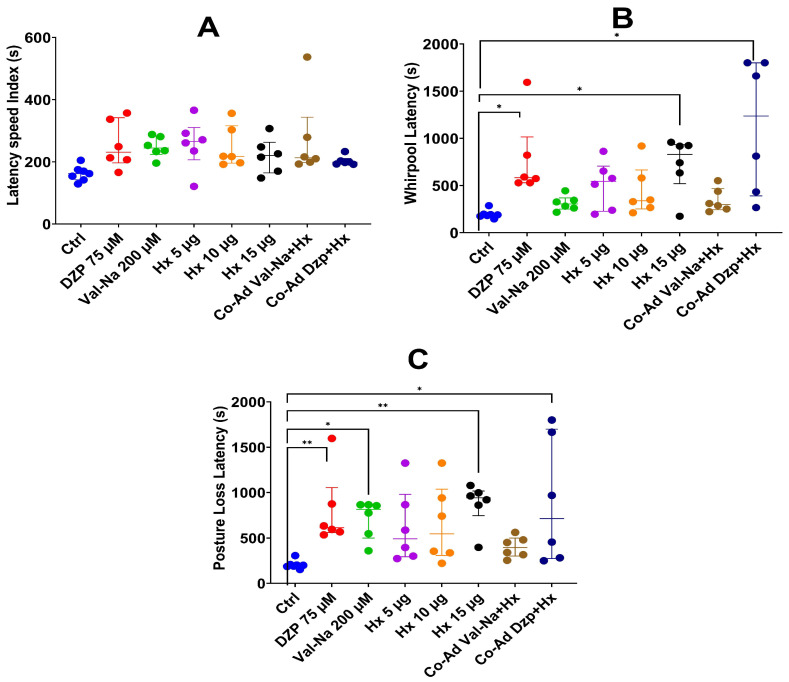
Latency metrics for zebrafish treated with *Verbesina persicifolia* hexane extract in combination with pharmacological controls. (**A**) Speed Index Latency, (**B**) Whirlpool Latency, and (**C**) Posture Loss. The administration of sodium valproate and the hexanic partition was at a concentration of 200 µM and 10 µg/mL, respectively. The administration of diazepam and the hexanic partition was at a concentration of 37.5 µM and 5 µg/mL, respectively. (**B**) (* *p* = < 0.01); (**C**) (* *p* = 0.01), (** *p* = < 0.01). Data are presented as medians ± interquartile ranges. For A, the analysis was performed using ANOVA followed by Tukey HSD, while for B and C, Kruskal–Wallis was used, followed by the Bonferroni correction for the Mann–Whitney U test.

**Figure 4 plants-14-01078-f004:**
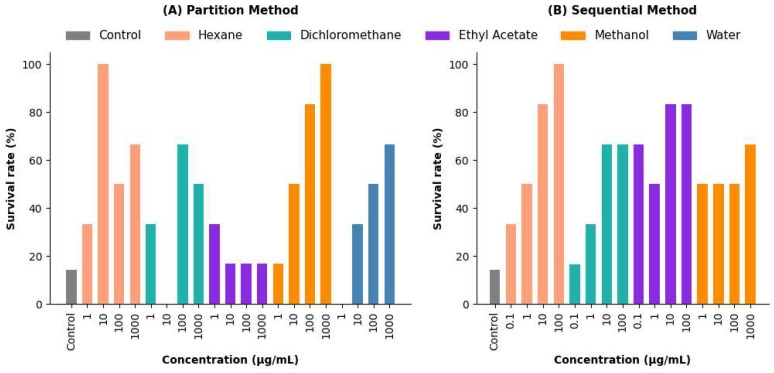
Survival rates (%) of *Verbesina persicifolia* extracts in zebrafish under different extraction methods. (**A**) Partition method: survival rates for hexane, dichloromethane, ethyl acetate, methanol, and water extracts across concentrations (1, 10, 100, and 1000 µg/mL). (**B**) Sequential method: survival rates for the same solvents across concentrations (0.1, 1, 10, 100, and 1000 µg/mL).

**Table 1 plants-14-01078-t001:** Phytochemical screening of *V. persicifolia* methanolic extracts from 2019, 2020, and 2021.

Metabolite Family	2019 Extract	2020 Extract	2021 Extract
Alkaloids	+	+	+
Saponins	+	−	+
Tannins	−	+	+
Quinones	+	−	−
Triterpenoids	−	+	+
Flavonoids	+	+	+
Steroids	+	+	+

(+): positive results, (−) negative results.

**Table 2 plants-14-01078-t002:** Phytochemicals identified in *Verbesina persicifolia* based on the literature.

Metabolite Family	Extraction Method	Literature Source
Eudesmane sesquiterpenes	Hexane, methanol	[26]
Flavonoids	Ethyl acetate, methanol	[18]
Alkaloids	Methanol	[18]
Steroids	Hexane	[18]
Saponins	Methanol	[18]
Tannins	Methanol, aqueous extracts	[18]
Essential oils	Hexane	[27]

**Table 3 plants-14-01078-t003:** Toxicity assessment of *Verbesina persicifolia* methanolic partitions and sequential extracts in *Artemia salina*.

Extract Type	Fraction	LC_50_ (mg/mL)	Toxicity Classification ^1^
Methanolic extract partition	Hexane	0.499	Low toxicity
	Dichloromethane	>1	Non-toxic
	Ethyl acetate	0.633	Low toxicity
	Methanol	>1	Non-toxic
	Aqueous	>1	Non-toxic
Sequential extract	Hexane	0.073	High toxicity
	Dichloromethane	0.105	Moderate toxicity
	Ethyl acetate	0.421	Moderate toxicity
	Methanol	0.421	Moderate toxicity

^1^ Toxicity classification is based on Clarkson’s toxicity criteria, where LC_50_ > 1 mg/mL is considered non-toxic, LC_50_ between 0.5 and 1 mg/mL indicates low toxicity, LC_50_ between 0.1 and 0.5 mg/mL denotes moderate toxicity, and LC_50_ < 0.1 mg/mL indicates high toxicity.

## Data Availability

The data supporting the findings of this study are included in the article and its Appendix A.

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
