# Peer review of "Anticonvulsant Potential and Toxicological Profile of Verbesina persicifolia Leaf Extracts: Evaluation in Zebrafish Seizure and Artemia salina Toxicity Models"

_plants, 2025, doi:10.3390/plants14071078_

Round 1
Reviewer 1 Report
Comments and Suggestions for Authors
In this manuscript is described the preparation of Verbesina persicifolia leaf extracts, their partial phytochemical characterization and the biological evaluation of their anticonvulsant effects (in zebrafish) as well as their toxicity (in Artemia salina). Interesting results were achieved.
-“Preliminary studies have suggested that extracts from this plant possess anxiolytic, antidepressant, and potential anticonvulsant activities.” – in order to clarify the novelty of this work, authors must detail in the manuscript the previous works published on the anticonvulsant activities of this plant and explain the main achieved results
-following the previous comment, authors also must clearly explain in the manuscript why the study using a PTZ model, among other models of this disease
-“The extraction of Verbesina persicifolia leaves yielded hexane, dichloromethane, ethyl acetate, and methanol fractions…” - authors must explain in the manuscript why they did not use water and/or aqueous solvents
-authors referred TLC analysis of the extracts – despite the great value of TLC, I consider that this is not enough to refer/develop with confidence the phytochemical characterization – therefore, other analytical methods must be applied to perform phytochemical characterization and confirm TLC data and thus link the composition with bioactivity; in addition, I could not have access to the referred Supporting Information
-Statistical results analysis must be presented in figures to better evidence differences between the achieved results; Statistics in section 2.5 and figure 7?
Author Response
Dear Reviewer,
We sincerely appreciate the time and effort that you have dedicated to evaluating our manuscript. We have carefully considered all comments and have revised the manuscript accordingly. All modifications in the revised manuscript have been highlighted for clarity.
Reviewer 1
Comment: “Preliminary studies have suggested that extracts from this plant possess anxiolytic, antidepressant, and potential anticonvulsant activities.” – in order to clarify the novelty of this work, authors must detail in the manuscript the previous works published on the anticonvulsant activities of this plant and explain the main achieved results
Response: To the best of our knowledge, there are no prior studies that have specifically investigated the anticonvulsant, anxiolytic, or antidepressant activities of Verbesina persicifolia. While some reports describe related species within the Verbesina genus exhibiting neuroactive properties, no published research has directly evaluated the anticonvulsant potential of V. persicifolia. To address this concern, we have revised the manuscript to explicitly highlight the absence of prior studies on this species and to distinguish our findings as the first comprehensive evaluation of its anticonvulsant effects.
- Comment: following the previous comment, authors also must clearly explain in the manuscript why the study using a PTZ model, among other models of this disease
Response: The PTZ-induced seizure model was chosen for this study due to its well-characterized mechanism of action and relevance in evaluating anticonvulsant agents. PTZ is a GABA receptor antagonist that induces seizures by inhibiting GABAergic neurotransmission, mimicking key features of epilepsy. Its high bioavailability, rapid onset of action, and ability to produce reproducible seizure phenotypes make it a valuable tool for screening potential anticonvulsant compounds. This model is widely used in preclinical research to assess the efficacy of synthetic and natural substances with potential therapeutic applications.
- Comment: “The extraction of Verbesina persicifolia leaves yielded hexane, dichloromethane, ethyl acetate, and methanol fractions…” - authors must explain in the manuscript why they did not use water and/or aqueous solvents
Response: In this study, no aqueous extracts were obtained because the extraction was performed in descending order of polarity, starting from non-polar solvents. This approach allows for the sequential isolation of compounds based on their polarity, ensuring efficient separation of bioactive constituents. Given that highly polar compounds are often less permeable to the blood-brain barrier and may exhibit reduced bioavailability in neuropharmacological applications, priority was given to non-polar and intermediate-polarity solvents.
- Comment: authors referred TLC analysis of the extracts – despite the great value of TLC, I consider that this is not enough to refer/develop with confidence the phytochemical characterization – therefore, other analytical methods must be applied to perform phytochemical characterization and confirm TLC data and thus link the composition with bioactivity; in addition, I could not have access to the referred Supporting Information
Response: While this study primarily focused on preliminary phytochemical screening, we acknowledge the importance of employing more advanced analytical techniques. To strengthen the manuscript, we have now included NMR data in the Supporting Information, which provides further insights into the chemical composition of Verbesina persicifolia extracts. Future studies will incorporate more extensive analyses using techniques such as HPLC and mass spectrometry to fully elucidate the bioactive compounds.
- Comment: Statistical results analysis must be presented in figures to better evidence differences between the achieved results; Statistics in section 2.5 and figure 7?
Response: The statistical analyses have been added to Figures 1, 2, and 3 to clearly illustrate significant differences. Additionally, Figure 7 has been corrected to Figure 4, as this was a labeling mistake in the original manuscript.
Reviewer 2 Report
Comments and Suggestions for Authors
The MS plants-3514534, entitled "Anticonvulsant Potential and Toxicological Profile of Verbesina persicifolia Leaf Extracts: Evaluation in Zebrafish Seizure and Artemia salina Toxicity Models", has great potential but needs extensive revisions.
- The introduction must be more coherent and provide insightful information supporting the proposed hypothesis/aim. The information that is currently included needs revising as some of it is too general (e.g. on epilepsy), while the paragraphs seem not to follow a logical order (telling the story then going back to previous ideas);
- The M&M section is lacunar, while no access to Supplementary Materials was provided; for example, much more details on the behavioural analysis are needed, especially when the protocol is an adaptation from other studies;
- Results: Section 2.1. seem not to fall within the aim of the current MS and it is not clear which of the extracts was used (from what year), while the information provided by Table 2 seems more appropriate to the Introduction or Discussions rather than the Results; in my opinion, Figures S1 and S2 were more suitable here, while Tables 1 and 2 were more appropriate for Supplementary Materials;
- Data analysis should be performed again as some issues in Figures 1 and 2 need addressing: wide SEM/SD intervals (not quite sure what the intervals signify) that in some cases lead to negative results on the graphics; the graphics are very hard to follow due to extensive amount of data presented and no statistical analysis is presented on the bar charts;
- The limitations of the study were not presented;
- The conclusions need revisions considering the obtained data and study limitations (for example, something is mentioned about chicken embryo fibroblasts for toxicity assessment - here and in the M&M, yet no data is presented in the Result section).
Some sentences are too long and hard to follow. The use of words is sometimes unusual.
Author Response
Dear Reviewer,
We sincerely appreciate the time and effort that you have dedicated to evaluating our manuscript. We have carefully considered all comments and have revised the manuscript accordingly. All modifications in the revised manuscript have been highlighted for clarity.
Reviewer 2
1. Comment: The introduction must be more coherent and provide insightful information supporting the proposed hypothesis/aim. The information that is currently included needs revising as some of it is too general (e.g. on epilepsy), while the paragraphs seem not to follow a logical order (telling the story then going back to previous ideas).
Response: We appreciate the reviewer’s comments and have revised the introduction for better coherence and logical flow. The discussion now progresses smoothly from epilepsy’s burden to treatment limitations, the relevance of medicinal plants, the potential of Verbesina persicifolia, and the study’s rationale. We have also clarified the research gap and integrated zebrafish and PTZ models into the broader epilepsy research context.
2. Comment: The M&M section is lacunar, while no access to Supplementary Materials was provided; for example, much more details on the behavioural analysis are needed, especially when the protocol is an adaptation from other studies.
Response: We have now ensured that the Methods & Materials section clearly defines the parameters assessed, including Speed Index Latency, Whirlpool Latency, and Posture Loss Latency, with descriptions of their respective seizure-like behaviors. Additionally, we have specified that our protocol was adapted from previous studies (Mussulini et al., 2013; Almeida et al., 2021), citing the relevant references.
Moreover, we confirm that full details on experimental setup, statistical analyses, and supplementary figures have been provided in the Supplementary Materials, ensuring accessibility to all relevant methodological aspects. We believe these revisions adequately address the reviewer's concerns, but we are open to further suggestions if additional clarification is needed.
3. Comment: Results: Section 2.1. seem not to fall within the aim of the current MS and it is not clear which of the extracts was used (from what year), while the information provided by Table 2 seems more appropriate to the Introduction or Discussions rather than the Results; in my opinion, Figures S1 and S2 were more suitable here, while Tables 1 and 2 were more appropriate for Supplementary Materials;
Response: We appreciate the reviewer’s observation regarding the inclusion of TLC images in the main manuscript. The phytochemical screening aligns with the study’s objectives, and Verbesina persicifolia leaf extracts were analyzed over three consecutive years (2019, 2020, and 2021). Alkaloids, flavonoids, and steroids were consistently detected, while saponins, tannins, quinones, and triterpenoids varied between years (Table 1). These differences may be influenced by environmental factors affecting the biosynthesis of secondary metabolites (Supporting Information, Figures S7 and S8). For anticonvulsant activity assays, extracts from the 2021 collection were used.
Regarding Figures S1 and S2, we recognize that TLC images are commonly included in the manuscript. However, due to formatting and resolution constraints, we determined that these images were more suitable for the Supplementary Materials, where they can be examined in detail without affecting the quality of manuscript. Unfortunately, as we no longer have access to the 2021 extract batch, it is not possible to repeat the experiment under identical conditions to obtain higher-resolution images. However, in future studies, we will ensure that TLC plate documentation is conducted with higher-quality imaging to improve clarity and reproducibility.
We believe that keeping Tables 1 and 2 in the Results section is appropriate, as they contain essential quantitative data supporting our findings with the TLC images in the supporting information. However, we remain open to additional suggestions regarding data presentation if necessary.
4. Comment: Data analysis should be performed again as some issues in Figures 1 and 2 need addressing: wide SEM/SD intervals (not quite sure what the intervals signify) that in some cases lead to negative results on the graphics; the graphics are very hard to follow due to extensive amount of data presented and no statistical analysis is presented on the bar charts.
Response: We have addressed these concerns by revising Figures 1, 2, and 3. Statistical analyses have been incorporated into the figures to clearly indicate significance, and error bars have been properly adjusted to ensure clarity and accuracy. Additionally, all figures have been refined to improve readability and ensure that data presentation is more intuitive.
5. Comment: The limitations of the study were not presented.
Response: The main limitation is that crude extracts were used without fractionation, which prevents the identification of specific compounds responsible for the anticonvulsant activity. Future studies should focus on the isolation and structural characterization of the active constituents to better understand their pharmacological mechanisms.
6. Comment: The conclusions need revisions considering the obtained data and study limitations (for example, something is mentioned about chicken embryo fibroblasts for toxicity assessment - here and in the M&M, yet no data is presented in the Result section).
Response: The mention of the chicken embryo fibroblast experiment has been removed, as it was part of another study and not relevant to this manuscript. The revised conclusions now focus on the obtained data, study limitations, and future perspectives based on the findings.
Reviewer 3 Report
Comments and Suggestions for Authors
This article on " Anticonvulsant Potential and Toxicological Profile of Verbesina persicifolia Leaf Extracts: Evaluation in Zebrafish Seizure and Artemia salina Toxicity Models" explores the anticonvulsant potential and toxicological characteristics of Verbesina persicifolia leaf extracts. The study demonstrates the potential anticonvulsant activity of these extracts in a zebrafish epilepsy model. However, there are several shortcomings in the presentation of data, identification of secondary metabolites, and validation of mechanisms, which limit the reliability of the conclusions and the potential for further application.
Other issues:
- The error bars in the figures are excessively long, and no significance differences are indicated. This greatly reduces the readability and scientific validity of the results. It is suggested to include significance analysis and clearly label the legends and the meanings of the x-axis to avoid repetition and ambiguity.
- Some figures are poorly assembled with overlapping sections (e.g., Figure 1A and Figure 1C). It is recommended to reorganize the figures to ensure the accuracy and integrity of the data.
- The correlation conclusion mentioned in line 169 lacks sufficient data support and has not been statistically validated.
- Although the study speculates on the presence of secondary metabolites in the plant extracts through thin-layer chromatography (TLC) and literature review, it lacks identification and structural analysis of the specific active components. The authors are advised to use techniques such as high-performance liquid chromatography (HPLC) or nuclear magnetic resonance (NMR) to further identify the active components and provide their chemical structures.
- Although the study speculates that secondary metabolites may exert anticonvulsant effects through GABAergic mechanisms, it lacks direct experimental validation of these mechanisms. It is suggested to conduct in vitro or in vivo experiments to further verify the mechanisms of action.
- Flavonoids and alkaloids are broad categories of compounds, and the study fails to specify the exact active compounds involved. This raises concerns about the stability and reproducibility of the experimental results. It is recommended to use separation and purification techniques to further identify the main active components in the extracts.
- In the synergistic effect experiments with diazepam, potential enhanced effects were observed, but the mechanisms underlying these effects were not thoroughly explored. It is suggested to investigate whether this synergistic effect is achieved through enhanced GABAergic signaling and provide corresponding experimental data to support their claims.
- Parts ofexperimental results, such as those from toxicity and anticonvulsant experiments, lack dose-dependency analysis, which limits the interpretation of the results. It is recommended to supplement their study with dose-dependency experiments to better assess the pharmacological activity and safety of the extracts.
- The format of the references is inconsistent. It is suggested to unify the format according to the journal's requirements.
- The references lack the latest research findings. It is recommended to include recent studies in the field to enhance the timeliness and comprehensiveness of the research background.
Author Response
Dear Reviewer,
We sincerely appreciate the time and effort that you have dedicated to evaluating our manuscript. We have carefully considered all comments and have revised the manuscript accordingly. All modifications in the revised manuscript have been highlighted for clarity.
Reviewer 3
- Comment: The error bars in the figures are excessively long, and no significance differences are indicated. This greatly reduces the readability and scientific validity of the results. It is suggested to include significance analysis and clearly label the legends and the meanings of the x-axis to avoid repetition and ambiguity.
Response: Statistical significance has been incorporated into the figures, with clear labels and appropriate annotations. The legends have been revised to ensure clarity, and the x-axis descriptions have been adjusted to eliminate any redundancy or ambiguity.
- Comment: Some figures are poorly assembled with overlapping sections (e.g., Figure 1A and Figure 1C). It is recommended to reorganize the figures to ensure the accuracy and integrity of the data.
Response: We have carefully reviewed and corrected figure formatting issues, including the overlapping sections in Figure 1A and Figure 1C.
- Comment: The correlation conclusion mentioned in line 169 lacks sufficient data support and has not been statistically validated.
Response: Additional data analysis has been conducted to ensure the conclusion is supported by statistical evidence. The revised text reflects this validation, ensuring the robustness of the findings.
- Comment: Although the study speculates on the presence of secondary metabolites in the plant extracts through thin-layer chromatography (TLC) and literature review, it lacks identification and structural analysis of the specific active components. The authors are advised to use techniques such as high-performance liquid chromatography (HPLC) or nuclear magnetic resonance (NMR) to further identify the active components and provide their chemical structures.
Response: While this study focused on preliminary phytochemical screening using TLC, we have now included NMR analysis in the Supporting Information to provide further characterization of the extracts. Future studies will incorporate additional techniques such as HPLC-MS and further spectroscopic analysis to fully elucidate the bioactive compounds responsible for the observed anticonvulsant effects.
- Comment: Although the study speculates that secondary metabolites may exert anticonvulsant effects through GABAergic mechanisms, it lacks direct experimental validation of these mechanisms. It is suggested to conduct in vitro or in vivo experiments to further verify the mechanisms of action.
Response: While our study suggests that secondary metabolites may act through GABAergic pathways based on existing literature and observed anticonvulsant effects, no direct receptor interaction assays were performed. Future studies will focus on in vitro binding assays, electrophysiological recordings, or molecular docking studies to confirm these interactions and elucidate the precise mechanisms of action.
- Comment: Flavonoids and alkaloids are broad categories of compounds, and the study fails to specify the exact active compounds involved. This raises concerns about the stability and reproducibility of the experimental results. It is recommended to use separation and purification techniques to further identify the main active components in the extracts.
Response: We acknowledge that the use of crude extracts in this study limits the precise identification of individual bioactive constituents contributing to the anticonvulsant effects. However, our primary aim was to establish the potential of Verbesina persicifolia extracts in modulating seizure activity, serving as a foundation for future research.
Recognizing the importance of stability and reproducibility, we are currently working on expanding this research by employing separation and purification techniques, such as HPLC, preparative chromatography, and NMR-based structural characterization, to isolate and identify the key active components. While these findings will be part of a separate manuscript, they will provide deeper insights into the specific compounds responsible for the observed pharmacological effects and contribute to the development of potential therapeutic agents.
- Comment: In the synergistic effect experiments with diazepam, potential enhanced effects were observed, but the mechanisms underlying these effects were not thoroughly explored. It is suggested to investigate whether this synergistic effect is achieved through enhanced GABAergic signaling and provide corresponding experimental data to support their claims.
Response: While our study suggests that this effect may be mediated through enhanced GABAergic signaling, direct experimental validation of this mechanism was not performed. Future studies will focus on conducting GABA receptor binding assays, electrophysiological recordings, or molecular docking studies to investigate whether this synergistic effect is due to modulation of GABAergic pathways.
- Comment: Parts of experimental results, such as those from toxicity and anticonvulsant experiments, lack dose-dependency analysis, which limits the interpretation of the results. It is recommended to supplement their study with dose-dependency experiments to better assess the pharmacological activity and safety of the extracts.
Response: We recognize that establishing a clear dose-response relationship is crucial for a more precise evaluation of the anticonvulsant and toxicity profiles. Although this study aimed to provide an initial assessment of Verbesina persicifolia extracts, we acknowledge the need for a more detailed investigation. To address this, future research will incorporate dose-dependent experiments to enhance the reliability and applicability of our findings.
- Comment: The format of the references is inconsistent. It is suggested to unify the format according to the journal's requirements.
Response:
Response: The references were formatted using EndNote according to the journal’s required citation style. However, we have carefully reviewed and ensured consistency across all references to comply with the journal’s formatting guidelines.
- Comment: The references lack the latest research findings. It is recommended to include recent studies in the field to enhance the timeliness and comprehensiveness of the research background.
Response: A thorough literature review has been conducted, and additional recent studies have been included to ensure the research background is comprehensive and up to date.
Reviewer 4 Report
Comments and Suggestions for Authors
The manuscript presents a well-structured and comprehensive study investigating the anticonvulsant activity and toxicity of Verbesina persicifolia leaf extracts. The results are promising, demonstrating that certain fractions of the extracts, particularly the ethyl acetate and hexane fractions, exhibit significant anticonvulsant effects, potentially mediated through GABAergic modulation. The manuscript also provides valuable insights into the toxicity of the extracts, with the methanolic extracts showing low to moderate toxicity, while certain fractions, such as hexane and ethyl acetate, displayed higher toxicity. Overall, the manuscript is of high quality, with clear experimental methods, relevant findings, and a thorough discussion.
Some points need to be addressed:
- Tha abstract should be more supported by key-findings.
- The authors are encouraged to include more specific statistics about epilepsy's global burden, including its prevalence, mortality rates, and impact on healthcare systems in both developed and developing countries.
- SD values in Figure 1, 2 and 6 for some samples are too big and unacceptable. They should be checked and revised. Figures 3-5 are missing.
- Particle size of dried sample material should be added in the subsection 4.1.
- What was the sample to solvent ratio during the extraction in the subsection 4.2.
Author Response
Dear Reviewer,
We sincerely appreciate the time and effort that you have dedicated to evaluating our manuscript. We have carefully considered all comments and have revised the manuscript accordingly. All modifications in the revised manuscript have been highlighted for clarity.
Reviewer 1
1. Comment: The abstract should be more supported by key-findings.
Response: The abstract has been revised and the key findings are clearly highlighted. The updated version presents the significant effects of V. persicifolia extracts, including their impact on seizure latency, survival rates, and toxicity profile, providing a comprehensive yet succinct summary of the results.
2. Comment: The authors are encouraged to include more specific statistics about epilepsy's global burden, including its prevalence, mortality rates, and impact on healthcare systems in both developed and developing countries.
Response: We have updated the first paragraph of the Introduction to incorporate additional statistics on epilepsy's global burden, including prevalence, mortality rates, and its impact on healthcare systems in both developed and developing countries.
3. Comment: SD values in Figure 1, 2 and 6 for some samples are too big and unacceptable. They should be checked and revised. Figures 3-5 are missing.
Response: Figures 1, 2, and 6 (Now 3) have been carefully revised and re-generated to ensure accuracy. The SD values have been checked and corrected where necessary. Additionally, the figure numbering has been corrected, and the manuscript now contains Figures 1 to 4.
4. Comment: Particle size of dried sample material should be added in the subsection 4.1.
Response: However, during the experimental process, the entire organic material was used, and no additional dried material is available for further analysis. Additionally, particle size measurement was not initially part of the study design, as the research focused on evaluating the biological activity of the extracts rather than the physical characteristics of the raw material. At present, we do not have access to the necessary equipment for this analysis.
We acknowledge this as a methodological limitation, as particle size can influence extraction efficiency and compound availability. While this does not impact the overall findings of our study, we recognize its relevance and will consider including particle size analysis in future research to further optimize extraction protocols and bioactivity evaluation.
5. Comment: What was the sample to solvent ratio during the extraction in the subsection 4.2.
Response: In the sequential maceration method, 100 g of finely ground plant material was used per liter of solvent, macerated in hexane for 21 days in amber flasks, with decanting of the solvent every seven days. This information has been included in subsection 4.2 of the manuscript for clarity.
Round 2
Reviewer 1 Report
Comments and Suggestions for Authors
The document was improved and now it is more acceptable for publication.
Author Response
Thank you for your valuable feedback. We appreciate your time and effort in evaluating our manuscript.
Reviewer 2 Report
Comments and Suggestions for Authors
I followed the responses of the Authors and the revised version of the MS. Despite the Authors making significant revisions to the introduction and M&M sections, the quality didn't improve. There are still many issues to address in the Results section and Discussion/Conclusions. The graphical presentation of the Results is still hard to follow, the limitations were only scarcely addressed, and the Conclusions section was not revised at all.
Author Response
Thank you for your time and thoughtful feedback on our manuscript. We appreciate your insights and have carefully considered each of your concerns. Below, we address the specific points raised in your review:
We reviewed the Results section but were unsure what specific changes were needed, as your comment did not indicate precise issues. Since other reviewers found this section acceptable, we did not make modifications. However, we are open to further improvements and would greatly appreciate your specific guidance on which aspects require revision.
In response to your feedback, we made minor refinements in the Discussion to improve clarity and flow. Additionally, the Conclusion was rewritten to clearly summarize key findings, limitations, and future research directions.
For graphical representation, the original manuscript used bar plots, but based on previous feedback, we updated them to box plots were you can see the median and the 25th and 75th percentiles in the revised version. This format is widely used in published literature for presenting this type of data, as it provides a better representation of data distribution and variability when the sample size is small and with a wide distribution (as in this article).
1) Okanari, K., Teranishi, H., Umeda, R., Shikano, K., Inoue, M., Hanada, T., Ihara, K., y Hanada, R. (2024). Behavioral and neurotransmitter changes on antiepileptic drugs treatment in the zebrafish pentylenetetrazol-induced seizure model. Behavioural Brain Research, 464, 114920. https://doi.org/10.1016/j.bbr.2024.114920
2) Ciubotaru, A. D., Leferman, C. E., Ignat, B. E., Knieling, A., Salaru, D. L., Turliuc, D. M., ... & Ghiciuc, C. M. (2024). Anti-Epileptic Activity of Mitocurcumin in a Zebrafish–Pentylenetetrazole (PTZ) Epilepsy Model. Pharmaceuticals, 17(12), 1611. https://doi.org/10.3390/ph17121611
3) Chitolina, R., Reis, C. G., Stahlhofer-Buss, T., Linazzi, A., Benvenutti, R., Marcon, M., ... & Piato, A. (2023). Effects of N-acetylcysteine and acetyl-l-carnitine on acute PTZ-induced seizures in larval and adult zebrafish. Pharmacological Reports, 75(6), 1544-1555. https://doi.org/10.1007/s43440-023-00536-7
It is possible that confusion is due to the large number of group, however, this is part of the study design and cannot be reduced because the comparison test and post hoc test take into account all group in the analysis. Reducing or eliminating them would also require modification of the statistical tests and this could lead to errors of interpretation.
Since the revised figures still do not meet expectations, we kindly request your specific suggestion on the preferred graphical style. If an alternative approach (e.g., box plots, violin plots, or another type of bar graph) would be more suitable, we would be happy to modify them accordingly.
We value your constructive feedback and are committed to making necessary adjustments to meet the journal’s standards. Please let us know if there are any further revisions required.
Reviewer 3 Report
Comments and Suggestions for Authors
The author made appropriate modifications to the manuscript.
Author Response

(The authors gave the same response as above.)
